# OpenReview forum: "SynQuE: Estimating Synthetic Dataset Quality Without Annotations"
_TMLR — Accepted by TMLR_

### Review · Reviewer_GLM5 · 2026-02-27

**Summary Of Contributions:**

This paper introduces and formalizes the problem, which the authors refer to as Synthetic Dataset Quality Estimation (SYNQUE). Specifically, the general idea of this proposed task is to evaluate and select the “best” synthetic dataset that shall be most beneficial to downstream tasks when trained on the synthetic dataset, and evaluated on real test data, without the necessity to access labeled validation sets.

Overall speaking, the authors claim their three main contributions to be: (1) the formalization of the problem and the introduction of the benchmark. (2) An LLM-based proxy metric that can be used for the dataset selection. (3) Empirical validation across 4 tasks covering sentiment analysis, Text2SQL, web navigation, and image classification.

**Audience:**

Yes

**Audience Explanation:**

The problem and motivation are quite interesting and relevant.

**Broader Impact Concerns:**

From my perspective, there are potential ethical implications. Since the proposed task and selection process are based purely on test performance on real-world data, this may amplify known biases present in the synthetic datasets.

**Claims And Evidence:**

No

**Claims Explanation:**

Although I find the high-level motivation and the idea of finding a more efficient way to select synthetic data that maximally benefits downstream real-world application performance interesting and useful, I have multiple concerns regarding the current statements and experimental setup.

1. The paper claims that the proposed method is a dataset-level ranking mechanism, and thus different from previous instance-level selection techniques. However, I don’t think there is an actual technical bottleneck preventing moving between these two levels. In other words, one should be able to apply instance-level selection techniques to each individual data sample in a dataset and then aggregate over them to select the “best” dataset; similarly, the proposed method could be applied to smaller datasets with just a few samples. That being said, it would strengthen the work if the authors could incorporate such instance-level selection methods as baselines to provide a more comprehensive comparison.

2. I am also concerned about the proposed suite of metrics, including the LLM-based LENS. As the authors mention, synthetic data may contain hallucinations, which can ultimately hinder downstream task performance when such data are included in training. I was expecting the proposed method to take this into account, but the current selection mechanism does not seem to explicitly address this inherent limitation of synthetic datasets.

3. The observed performance variations and correlation patterns are rather mixed. The language-based tasks seem to benefit more than image classification, where adding synthetic datasets can actually hurt performance. This raises questions about whether the proposed task is truly valuable for vision applications.

**Requested Changes:**

In addition to the points mentioned above, I have the following suggestions for improvement:

1. While the “theoretical” targeted downstream tasks that will benefit from this selection mechanism are those applications where real-world data is sparse and difficult to collect, the downstream applications selected for empirical validation include more commonly studied tasks where data sparsity is not a major challenge. It would strengthen the paper if the authors could demonstrate effectiveness on the truly targeted tasks they claim.

2. It would be necessary to report the actual computational cost and runtime for each metric, especially the proposed LENS. I get the impression that leveraging multiple LLMs for reasoning and judgment is not a cheap process. If the selection process itself becomes computationally expensive, it would be difficult to convince the community of its practical utility.

3. The current synthetic datasets are all rather small, each containing roughly 1000 samples. It is unclear whether the proposed method scales to larger datasets or how computationally expensive this would be. It would be beneficial to evaluate the robustness of these proxy metrics with respect to dataset scale.

4. This scale bias also affects the proxy metrics, particularly MDM, which measures diversity. Intuitively, larger datasets may benefit from broader diversity coverage, so I wonder whether there are any assumptions or constraints regarding the synthetic dataset sizes among which we aim to select.

---

> ### Author Response · Authors · 2026-03-27
>
> We thank the reviewer for their thoughtful feedback and for finding the problem and motivation `"quite interesting and relevant"`. We appreciate the constructive questions, which have guided us to add significant new ablations and analyses that strengthen the paper. We address your concerns below:
>
> **[Instance-Level Baselines]**
>
> The reviewer raises an insightful point regarding instance-level selection. While scaling instance-level selection across all datasets and domains via LLMs is computationally expensive given limited time budget, we agree that it is an important baseline.
>
> To address this, we conducted an ablation study on the Sentiment Analysis task, which has a sufficiently large pool of samples. We merged all synthetic datasets, used each metric to select a class-balanced subset of the top 999 synthetic samples, and evaluated the test accuracy of a regression model trained on these samples.
>
> The results are as follows, using the format of (dataset-level/instance-level) (delta in accuracy):
> - PAD (55.3/55.9) (+0.6)
> - Debiased LENS (Qwen2.5-7B-Instruct) (50.5/54.8) (+4.3)
> - MMD² (54.7/53.2) (-1.5)
> - MAUVE (54.6/52.7) (-1.9)
>
> Crucially, while instance-level selection strongly outperforms the baseline task average (49.6%), our results reveal a critical scientific insight that justifies the necessity of dataset-level evaluation:
> While greedy instance-level selection slightly improves discriminative proxies (PAD: +0.6, LENS: +4.3), it actually degrades the performance of generative and distributional metrics (MMD² drops by -1.5, MAUVE drops by -1.9). By greedily selecting only the "safest" individual samples, instance-level filtering reduces the natural diversity and coverage of the dataset, inducing mode collapse. Therefore, holistic dataset-level evaluation (SynQuE) is necessary to preserve distribution geometry. Furthermore, dataset-level ranking ($O(K_{datasets})$) is vastly more scalable than evaluating millions of individual samples across candidate pools ($O(N_{total})$) during the bulk data procurement phase. We have added these baseline results and this trade-off discussion to the revised Appendix (B.12).
>
>
> **[Handling Hallucinations]**
>
> We thank the reviewer for raising this critical point. We agree that generative model hallucinations are an inherent challenge in synthetic data that can harm downstream performance [R1]. Detecting these factual inconsistencies without access to ground-truth labels remains a difficult open problem. To rigorously explore this limitation and test whether **LENS** could explicitly address it, we conducted two new sets of experiments on the Sentiment Analysis task, which we have now added to the Appendix (B.10):
> - **Explicit Filtering via LENS**: We explicitly instructed the LLM scorer to identify and penalize hallucinated examples during the scoring phase by adding a "de-hallucination prompt." We benchmarked this capability across multiple frontier models (Qwen2.5-7B-Instruct, Gemma-3-12B-It, Ministral-8B-Instruct-2410, and Granite3.3-8B-Instruct).
>   - **Results**: As shown in the newly added Table 16, the results are highly model-dependent. While explicit prompting with de-hallucination sentence yielded notable gains for some models (e.g., Ministral-8B saw its Spearman correlation improve from 0.28 to 0.53), it actually degraded the predictive power of others (e.g., Qwen2.5-7B Spearman dropped from 0.25 to 0.15). This underscores our position that relying purely on LLM prompting to perfectly isolate subtle logical errors is fragile and sensitive to the specific evaluator model used.
>
> While explicit hallucination filtering remains an imperfect science in a zero-shot, label-free setting, it is important to note that our representation-based proxies (such as MMD and PAD) *implicitly* penalize heavily hallucinated datasets. Because hallucinations introduce counterfactual artifacts, the resulting corrupted data distribution inherently diverges from the clean, real-world target distribution $\mathcal{U}_{r}$, resulting in a poorer SynQuE score. We have revised the manuscript to explicitly discuss these inherent limitations of synthetic data and have included the full ablation study detailing the de-hallucination prompting results in the Appendix (B.10).
>
> This mixed result highlights that explicit, zero-shot LLM de-hallucination is highly brittle and model-dependent. This is exactly why our framework relies on representation-based proxies (PAD, MMD²). These metrics implicitly penalize severe hallucinations without needing explicit prompt engineering. Because text corrupted by factual inconsistencies naturally diverges from the clean, real-world target distribution ($\mathcal{U}_r$) in the embedding space, it naturally receives a poorer, automatically penalized SynQuE score.

---

> > ### Author Response · Authors · 2026-03-27
> > **Cont. (1)**
> >
> > **[Cost Analysis and Runtime]**
> >
> > We completely agree that transparency regarding computational overhead is essential for demonstrating practical utility. To address this, we have added a comprehensive cost and runtime analysis for the Sentiment Analysis task (evaluating datasets of 999 samples) to the Appendix (B.11).
> > Specifically, we now report:
> > - **Token Usage**: The total number of input and output tokens required for LENS rubric compilation and scoring, which averages a highly manageable $9264 \pm 440$ tokens per dataset.
> > - **End-to-End Runtime**: The total end-to-end latency for LENS (benchmarking both Qwen2.5-7B and Qwen2.5-32B), alongside all representation-based metrics (PAD, MMD², MDM, and Mauve).
> > Our findings demonstrate that while the selection process does carry overhead, it is highly practical for offline dataset selection. While representation-based metrics like MMD² (6.12s) and PAD (8.07s) are the fastest, our LLM-based LENS-7B evaluator (90.52s) actually processes a dataset faster than the established generative baseline Mauve (112.51s). Furthermore, even our larger LENS-32B model evaluates an entire dataset in under five minutes (271.25s).
> >
> > We believe this explicit reporting clearly demonstrates to the community that LENS provides advanced reasoning capabilities while remaining highly competitive in throughput relative to existing generative evaluation methods.
> >
> > **[Vision Performance]**
> >
> > We note the mixed results in image classification, which are due to training vision models from scratch on small synthetic datasets without pre-training to *avoid data contamination*. While this makes learning robust features harder, this experiment highlights SynQuE's modality-agnostic nature. Recognizing synthetic datasets that harm performance is as important as finding those that help.
> >
> > **[Sparsity Justification]**
> >
> > We thank the reviewer for highlighting the importance of evaluating tasks that genuinely reflect our targeted use cases. Our primary goal was to establish the first comprehensive benchmark for the SynQuE problem, which required selecting testbeds that evaluate our baselines holistically across diverse modalities and task formats. However, we specifically selected tasks that do represent severe data scarcity and privacy constraints, alongside more standard modalities.
> > To clarify our selection rationale, which we will emphasize more strongly in the revised manuscript:
> > - **Demonstrated on Highly Targeted Tasks**:
> >   - **Web Navigation**: Agentic web navigation is currently bottlenecked by a severe lack of high-quality training trajectories in the wild (as noted by recent works [R2,R3]). Our evaluation in this domain explicitly demonstrates SynQuE's effectiveness on the exact type of costly, complex, and data-scarce problem we target.
> >   - **Text2SQL**: While standard SQL datasets exist, real-world applications such as enterprise Text2SQL represent highly *privacy-sensitive* domains where collecting diverse, real-world query data is notoriously restricted. This makes zero-shot synthetic data selection highly relevant here.
> > - **Methodological Justification for Standard Modalities**:
> >   - **Sentiment Analysis**: To test fundamental stylistic and distributional shifts, we needed a classification task. However, to ensure rigorous evaluation, we deliberately selected a non-popular, domain-specific financial news dataset rather than standard benchmarks like the IMDB sentiment analysis dataset. This was to maximally avoid LLM data contamination, a major confounder in synthetic data research.
> >   - **Image Classification**: We included this to demonstrate that our representation-based proxies can extrapolate beyond text to visual modalities, evaluating on datasets with inherent visual ambiguity and label noise.
> >
> > Finally, while standard datasets for sentiment and image classification may not suffer from global data scarcity, our experimental setup artificially restricts the available real data to a tiny, unannotated subset ($\mathcal{U}_r$). We maintain that the fundamental principle of our framework—that minimizing distributional divergence bounds downstream model error—is domain-agnostic and translates directly to settings with absolute data scarcity.

---

> > > ### Author Response · Authors · 2026-03-27
> > > **Cont. (2)**
> > >
> > > **[Scale & Robustness of Proxies]**
> > >
> > > We thank the reviewer for this highly intuitive and critical observation. We completely agree that scale bias—particularly for diversity measures like MDM -- is an important factor, and we appreciate the opportunity to clarify the assumptions of our framework.
> > >
> > > To evaluate the robustness of our proxy metrics with respect to dataset scale, we conducted a new scaling ablation on the Sentiment Analysis task, which provides the most abundant synthetic training data (32 datasets of 998 samples each). Rather than downsampling, we systematically merged $n$ datasets together ($n \in \{1, 2, 4, 8\}$) to evaluate how effectively the proxies maintain their ranking capabilities as the synthetic dataset size increases up to roughly 8000 samples.
> > >
> > > Our findings, now detailed in the revised Appendix (B.11), reveal the following:
> > > - **Scale Degradation**: As you correctly hypothesized, increasing the synthetic dataset size generally leads to performance degradation across all proxies. Comparing datasets of vastly different scales introduces inherent biases (e.g., larger datasets naturally covering more embedding space), confirming the underlying assumption of the SynQuE framework: candidate datasets should be of comparable size to ensure fair quality estimation.
> > > - **Robustness of LENS**: Despite the general degradation, LENS demonstrated greater robustness to growing dataset sizes compared to the representation-based methods. It remained capable of achieving a positive correlation even when the size multiple reached $n=4$ (roughly 4000 samples).
> > > - **Rubric Sampling Constraints**: This experiment also highlighted a specific operational constraint for LENS. Throughout these scaling experiments, the number of synthetic samples drawn for rubric compilation was fixed at 200. As the total dataset size scales up, this fixed sample size represents a shrinking proportion of the data. This limits the LLM's ability to capture the full variance and nuance present in larger datasets within its rubric, which explains the eventual degradation at $n=8$.
> > >
> > > We have added these scaling results, alongside an explicit discussion of these size assumptions and rubric constraints, to the revised manuscript to better guide future applications of SynQuE.
> > >
> > > **[Broader Impacts]**
> > >
> > > We appreciate the reviewer raising this important ethical consideration, and we apologize if our initial phrasing regarding dataset selection caused ambiguity. To clarify, SynQuE **does not** select datasets by directly optimizing for downstream test performance (which is strictly inaccessible during the selection phase). Rather, methods like PAD, MMD², and MAUVE select synthetic data based strictly on its unsupervised distributional similarity to the small real-world reference set. We only use test performance as a post-hoc oracle to evaluate the success of our proxies.
> > > Therefore, the risk of amplifying biases depends entirely on the composition of the real-world anchor set. If the real reference set contains known biases, similarity-based selection will prioritize synthetic data that reflects those biases. We now explicitly discuss this mechanism in the "Broader Impacts Statement" section and advise practitioners (we added a "Practitioner Guidelines" section) to carefully audit their real-world reference samples for representational fairness before applying SynQuE.
> > >
> > > **References**:
> > >
> > > [R1] Gerstgrasser, M., et al. "Is Model Collapse Inevitable? Breaking the Curse of Recursion by Accumulating Real and Synthetic Data." COLM, 2024.
> > >
> > > [R2] Xu, Y., et al. "AgentTrek: Agent Trajectory Synthesis via Guiding Replay with Web Tutorials." ICLR, 2025.
> > >
> > > [R3] Murty, S., et al. "NNetNav: Unsupervised Learning of Browser Agents Through Environment Interaction in the Wild." SSI-FM Workshop at ICLR 2025, 2025.

---

> > > > ### Comment · Reviewer_GLM5 · 2026-04-10
> > > >
> > > > Thank the authors for the responses, my questions and concerns have been overall addressed. I suggest that the additional baselines and ablation should be included in the manuscript.

---

> > > > > ### Author Response · Authors · 2026-04-10
> > > > >
> > > > > We are pleased that our responses addressed your concerns and clarified your questions. Additional baselines and ablation studies have been incorporated into the revised manuscript. We sincerely thank you for your insightful feedback, which has helped strengthen the empirical depth of our work.

---

### Review · Reviewer_56e5 · 2026-03-04

**Summary Of Contributions:**

- The authors formalize the problem of Synthetic Dataset Quality Estimation, called SynQuE, where multiple synthetic datasets have to be ranked using only a small subset of unannotated real data which also represents the target distribution.
- To this end, the authors explore a suite of representation-based proxies such as mean distance to medoids (MDM), maximum mean discrepancy (MMD$^2$), proxy-a-distance (PAD) and Mauve. For long horizon settings where representations are unsuitable, the authors introduce LENS, a method that uses LLMs as zero-shot discriminators.
- The authors conduct evaluations on a comprehensive suite including sentiment analysis, Text-to-SQL, image classification, and web navigation tasks. This covers multiple domains, model sizes, model varieties, data modalities, etc.

Strengths:

1. The paper is overall well written and easy to read.
2. The paper addresses a very relevant problem today, about understanding and evaluating the quality of synthetic data for downstream tasks.
3. The explored solutions are fairly neat, and cover a range of different ways to tackle the problem. The reviewer particularly appreciates the systematic debiasing developed for LENS.
4. The experimental evaluations are quite comprehensive. The authors report both the improvement in test performance over uniform selection as well as the correlation between SynQuE metrics and test performance, offering a complete analysis of the proposed methods.

Weaknesses:
1. The notations used in Section 4.2 to describe the debiasing in LENS are difficult to follow and raise some ambiguities.
2. While LENS attains net positive results across all tasks, the performance improvements seem to be modest compared to representation-based methods such as Mauve on Sentiment and Text2SQL.
3. The improvements in image domain are generally small or negative.

**Audience:**

Yes

**Audience Explanation:**

Certainly, the problem is extremely relevant and the authors did a good job in identifying and evaluating promising solutions. I believe the findings of the paper would be relevant for a large subset of the TMLR community.

**Broader Impact Concerns:**

None.

**Claims And Evidence:**

Yes

**Claims Explanation:**

Yes, the claims are backed up via a comprehensive evaluation of all proposed methods. The results match the claims made. The experimental section demonstrates a thorough effort in justifying the claims.

**Requested Changes:**

The below changes are not critical to acceptance however would significantly improve the quality in the reviewer's opinion.

1. I would appreciate the inclusion of a small table that lists the number of synthetic datasets $(K)$ that were considered in each setup, against which the top-3 performance is compared. Furthermore, the information about size of real world inputs $(m_r)$, size of synthetic sets, size used for rubric evaluation, etc. is scattered in the text. This would be nice to also be included in the table for a quick overview.
2. Debiasing in LENS:
    - The meaning of four scoring permutations is unclear.
    - The quantity $h_{s\mid C_{s,r}}$ in Eq (6) is unclear. Could the authors provide an explicit formula for this?
    - In general, it is unclear in which part of the debiasing involves real samples vs. which part involves synthetic samples.
    - The definition of $g_{\mathcal{D}\mid C}$ can be more clearly stated, defining its input and output space.
3. Mean distance to Medoid: There is notational inconsistency.
    - What is $M$? In the beginning of Section 3, same letter is used for denoting the performance of model $f$.
    - Are there $N$ medoids or $N$ samples in each cluster? The aggregation of distances $\sum_i$ is computed over $N$.
4. Could the authors include more details on the test set used in the image experiments?
5. I would appreciate a small paragraph giving guidelines for practitioners, such as the recommended methods for each type of task (as it is understood that there is no single global winner). While the authors do provide such discussions scattered across the experiments, a summary would be nice.


Minor typos:

- Pg. 7 in Text2SQL: (...Store, Computer Students –– we the last two as Apps and Computers)
- Pg. 6 at the first line in Section 5: extrapolates should be plural
- Pg. 8 last line: Mentions that classes like "stage" in Split 1, however, "stage" is in Split 2 according to Table 4

---

> ### Author Response · Authors · 2026-03-27
>
> We thank the reviewer for their careful reading and detailed feedback. We are glad you found the problem relevant and the experimental evaluations comprehensive. We have formalized the mathematical notation in Section 4.2 and added the requested details and guidelines.
>
> **[W1 & C2. Notation Ambiguities in LENS]**
> We agree that the mathematical formulations in Section 4.2 required more explicit definitions. We have revised the manuscript to address these points directly:
> - **Input/Output Space**: We formalized the scoring function $g_{D|C} : \mathcal{X} \rightarrow \{0, 1, 2, 3, 4\}$, mapping an input sample to an integer corresponding to the LLM's likelihood assessment categories ({"very unlikely", "unlikely", "unsure", "likely", "very likely"}).
> - **The Four Scoring Permutations**: We explicitly state that evaluating a sample against two possible target labels ($D \in \{\text{real}, \text{synthetic}\}$) under two directional rubrics ($C \in \{C_{r,s}, C_{s,r}\}$) yields four raw scores per sample: $g_{r|C_{s,r}}(x)$, $g_{s|C_{s,r}}(x)$, $g_{r|C_{r,s}}(x)$, and $g_{s|C_{r,s}}(x)$.
> - **Explicit Formula for $h_{s|C_{s,r}}$**: We added the explicit formulation for the synthetic-target score-debiased function: $h_{s|C_{s,r}}(x) = \frac{g_{s|C_{s,r}}(x)}{\max(\epsilon, z_{s|C_{s,r}})}$, where the baseline $z_{s|C_{s,r}} = \frac{1}{n_r} \sum_{i=1}^{n_r} g_{s|C_{s,r}}(x_i)$.
> - **Sample Dependencies & Data Sources**: We updated Equations 5 and 6 to include the sample argument $(x)$ for the debiased functions $h(x)$ and $p(x)$ to clarify they are computed per sample. We also explicitly state that real samples ($\mathcal{U}_r$) are used exclusively to compute the normalization baselines ($z$), while the debiasing steps and final LENS score are applied strictly to the synthetic samples ($\mathcal{D}_s$) being evaluated.
> - **Indexing Correction**: We fixed the typographical error in Equation 8, correcting the denominator from $n$ to $n_s$ to accurately reflect the synthetic sample size.
>
> **[W2. Modest LENS Improvements on NLP Tasks]**
> We explicitly concede in the text that representation-based metrics like MAUVE and PAD are highly effective -- and sometimes superior -- for standard text tasks (like Sentiment Analysis), because dense continuous embeddings capture semantic features well. However, LENS is explicitly designed for complex, agentic, and long-horizon tasks (like Web Navigation). In these domains, standard embeddings struggle to capture the sequential logic required for success. By relying on zero-shot logical evaluation via LLM rubrics, LENS succeeds where representation metrics falter.
>
> Furthermore, our end-to-end benchmarking (Appendix C.4) demonstrates LENS remains highly practical: evaluating a typical 999-sample dataset takes \~90 seconds end-to-end on a vLLM server (with actual batched per-sample scoring inference taking \~15 seconds of active compute), which is faster than calculating MAUVE (~112 seconds) on the same hardware.
>
> **[W3 & C4. Vision Domain Improvements & Test Sets]**
> The modest improvements in the vision experiments stem from our strict experimental protocol: we train vision classifiers entirely from scratch (unpretrained ResNet) on small synthetic datasets. Utilizing pre-trained backbones (e.g., ImageNet weights) introduces pre-training data leakage that would confound the measured utility of the synthetic dataset itself. While this strict scientific control lowers absolute accuracy, it is necessary to isolate and measure true downstream utility. We have also updated Appendix C.1 to clarify that the image test sets consist of 150 label-balanced images per experimental split.
>
> **[C3. MDM Notational Inconsistency]**
> We have entirely revised the mathematical formulation for MDM to ensure precision. We removed the colliding variable $M$ and the ambiguous variable $N$. Now, we explicitly define $K$ as the total number of medoids, $n_s$ as the total number of synthetic samples, and $C_k$ as the subset of points assigned to the $k$-th medoid ($x_k$). The aggregation formula is now explicitly written as: $\mathrm{MDM} = \frac{1}{n_s} \sum_{k=1}^K \sum_{x_i \in C_k} d(x_i, \tilde{x}_k)$

---

> > ### Author Response · Authors · 2026-03-27
> > **Cont.**
> >
> > **[C1 & C5. Stats Tables and Practitioner Guidelines]**
> > We have added a consolidated table to the Appendix (C.3) summarizing: (1) the number of candidate synthetic datasets per setup, (2) the size of the real-world input sets, (3) the size of each synthetic dataset, and (4) the number of samples used for rubric evaluation.
> >
> > We also added a dedicated "Practitioner Guidelines" section to the manuscript, recommending:
> > - **One-shot classical NLP Tasks**: Representation-based matching (PAD, MAUVE) are highly effective and computationally efficient defaults.
> > - **Agentic/Sequential Tasks**: LENS is highly recommended as embeddings struggle to capture sequential logic.
> > - **Vision Tasks**: Distribution matching (MMD²) provides a solid baseline, but should be paired with explicit semantic filtering.
> > - **Mode Collapse**: MDM should be used universally as a secondary filter to ensure sufficient dataset coverage.

---

> > > ### Comment · Reviewer_56e5 · 2026-04-10
> > >
> > > I thank the authors for their comprehensive responses and updates. The changes to the notation make the LENS debiasing much clearer. I also appreciate the addition of practitioner guidelines, offering concrete recommendations.
> > >
> > > I am overall satisfied with the changes.

---

> > > > ### Author Response · Authors · 2026-04-10
> > > >
> > > > We thank the reviewer for their positive assessment. We are glad that the updated notation and the new practitioner guidelines improved the clarity and depth of our manuscript. We sincerely thank the reviewer for their valuable feedback throughout this process.

---

### Review · Reviewer_drNk · 2026-03-14

**Summary Of Contributions:**

This work studies the problem of ranking unannotated synthetic datasets based on the utility of models trained on these samples on real-world tasks. The challenges here are two-fold: the ranking is obtained (1) with very little unlabeled real dataset, and (2) without explicitly training large models on these synthetic datasets. To this end, the work considers several proxy scores derived from existing works to capture different properties of synthetic datasets. In addition to these proxy scores, the work also introduces an LLM-based score with the same objective.

**Audience:**

Yes

**Audience Explanation:**

This work is extremely relevant to the ML audience, as it studies how to choose synthetic datasets with the highest expected downstream utility.

**Claims And Evidence:**

Yes

**Claims Explanation:**

I understand the claim to be that these proxy scores can be used to rank synthetic datasets on their expected downstream utility. The authors may correct me if my understanding is wrong. This claim is only partially supported by the experimental results. In Table 3, most correlation numbers are less than 0.5, and many are close to zero (meaning "no correlation").

**Requested Changes:**

**C1. Related works**: The related works section needs to be rewritten to present the existing works on the synthetic dataset ranking problem. Right now, there are too many references in Sec. 2 without a clear message. I could not parse any pre-2024 works on synthetic dataset ranking from Sec. 2, although this question has been studied for a long time. Some relevant works may be found in the related works section of [R1].

**C2. Overall message of the paper**: The reason for choosing diversity and similarity-to-real in SynQuE is not explained. The chosen proxies are not connected to the objective in Eq. (1). Without that reasoning, I feel there is the following circular argument in the paper: SynQuE starts by choosing diversity (through MDM) and similarity-to-real (through MMD, PAD, and MAUVE) scores to rank synthetic datasets. Then, the results show that diversity and similarity-to-real have some positive (mostly < 0.5, but still > 0) correlation to the downstream performance (Tab. 3). So, what is the message of the paper? That diversity and similarity-to-real are useful indicators for downstream utility? If so, wasn't this conclusion also the intuitive reason for why they were used in SynQuE in the first place? If not, please explain why diversity and similarity-to-real were considered as the primary scores in SynQuE. I think this issue can be resolved with writing changes by appropriately grounding the intuitions in existing works.

**C3. Results**: The numbers are not consistent across tasks in Table 2, and the correlation scores are weak. Is it possible to plot the results as a scatter plot with a linear fit over the samples to visually see the trend?

**C4. Use of MAUVE**: The last paragraph of Sec. 4.1 starts by saying SynQuE uses "an adaptation of MAUVE" and ends by saying SynQuE uses "the MAUVE score directly." Given that MAUVE was designed to compare real distributions to synthetically generated distributions, it might be better to treat it as a baseline rather than a part of the proposed SynQuE framework.

**References**

[R1] Pillutla et al., "MAUVE Scores for Generative Models: Theory and Practice", JMLR 2023.

---

> ### Author Response · Authors · 2026-03-27
>
> We thank the reviewer for their constructive feedback. Your insights have helped us to better *contextualize* our empirical results and rigorously *ground our chosen metrics in established learning theory*. We address your specific concerns below.
>
> **[Partially Supported Claim & Correlation Scores]**
>
> We acknowledge that the global linear correlation coefficients in Table 3 are moderate. We have updated Section 5 to better contextualize these results: evaluating downstream utility for dataset selection is fundamentally a rank-order problem (identifying top candidates) rather than a task of perfect linear predictability. Because downstream neural network performance scales non-linearly with data quality, linear correlation metrics are highly sensitive to rank inversions among lower-performing datasets, which are of less practical concern.
>
> Successfully identifying the highest-utility datasets at the top of the distribution is the primary objective. As demonstrated in Table 2, selecting the top-3 datasets using SynQuE proxies consistently and significantly outperforms indiscriminate (uniform) selection. For example, in the Text2SQL Apps domain, top-3 selection raises accuracy from 30.4% to 38.4%.
>
> Furthermore, we have added context regarding the inherent evaluation variance in specific domains:
> - **Image Classification**: The datasets contain significant visual variability and label ambiguity (e.g., distinguishing between a "stage" and a "throne"), which naturally degrades correlation stability.
> - **Web Navigation**: As a complex, long-horizon agentic task, the utility of a trajectory depends on achieving a final goal over many steps, making it inherently difficult for standard representation-based metrics to evaluate smoothly.
>
> **[C1. Related Work & C4. Use of MAUVE]**
> Thank you for pointing out the need for clearer historical context. Specifically, we added a new subsection, "Dataset Ranking and Selection", which contrasts our work with two historical lines of research:
>
> - **Instance-Level Selection**: We discuss pre-2024 works focusing on subset curation, such as submodular data selection (Kaushal et al., 2018) and active learning (BatchBALD; Kirsch et al., 2019), to clarify the fundamental difference between curating individual samples and ranking entire datasets.
>
> - **Macro-Level Real Data Retrieval**: We introduce previous work on dataset search and retrieval (Chapman et al., 2020; Nadas et al., 2025) to highlight how traditional dataset ranking optimizes for search relevance among real-world data silos, rather than predicting downstream training utility for synthetic data under strict label scarcity.
>
> This isolates the novel problem SynQuE solves. Specifically regarding Pillutla et al. (2023), we now explicitly state that while MAUVE elegantly measures information divergence to assess unconditional generation fidelity, it was not inherently designed to predict a dataset's downstream training utility. Following your suggestion (C4), we have revised Section 4.1 to formally treat MAUVE as an established generative baseline, rather than a proposed component of the SynQuE framework.
>
> **[C2. Overall Message & Theoretical Grounding]**
> You raise an important point regarding the potential circularity of relying on intuition to select our proxies. We have revised Section 4 to explicitly connect our chosen proxies to the objective in Equation 1 by grounding them in established machine learning theory:
> - **Similarity (Distribution Matching)**: We explicitly ground Proxy-A-Distance (PAD) in domain adaptation theory from Ben-David et al. (2006). The theory proves that downstream target risk is strictly upper-bounded by the source training risk plus the domain divergence (measured by PAD). Because minimizing target risk on the real distribution is exactly our downstream utility objective (Eq. 1), ranking synthetic datasets by PAD mathematically targets the divergence term within the theoretical upper bound of the downstream error.
> - **Diversity**: We ground Mean Distance to Medoid (MDM) in embedding space coverage. If a synthetic dataset lacks diversity (e.g., mode collapse), its empirical support is artificially restricted. A model trained on this limited support will fail to generalize to the broader real-world target distribution, directly increasing the target risk defined in Equation 1.
>
> **[C3. Results & Scatter Plots]**
> We have added scatter plots with lines of best fit to the Appendix (C.4) for the Sentiment Analysis, Text2SQL, and Image Classification tasks to visualize these trends. We opted to exclude the global scatter plot for Web Navigation because the data spans 13 distinct categorical domains with highly clustered subsets.

---

> > ### Comment · Reviewer_drNk · 2026-03-31
> > **Response from reviewer drNk**
> >
> > I thank the authors for their response. Their response has largely addressed my concerns.
> >
> > With respect to their response to C2, I appreciate grounding PAD using prior works. However, the response about diversity metrics is the same as that in the paper. There is an empirical observation that poor data diversity leads to poor performance. This observation, which motivates the choice of using diversity to estimate synthetic data utility, is also the conclusion, isn't it? If that's the case, the work does not add any new knowledge regarding synthetic data utility prediction.
> >
> > One way to resolve this issue is to see if any combination of the metrics is a more informative metric (better than the sum of its parts). This way, the current work adds to the existing knowledge about the role of diversity (or any metric for that matter) in downstream utility. Another suggestion would be to dissect diversity into various sub-metrics, each measuring a different kind of diversity. That also would be a new thing that the readers can learn from the papers.

---

> > > ### Author Response · Authors · 2026-04-02
> > >
> > > Thank you for the thoughtful follow-up and for pushing us to extract deeper insights from our framework. We are glad that our previous response largely addressed your concerns regarding PAD.
> > >
> > > You raise a great point regarding the diversity metrics. We agree that while the principle that diversity aids generalization is established, the interaction between diversity and other distributional properties in synthetic data requires a deeper quantitative synthesis. Following your suggestion to investigate if a combination of metrics is more informative than the sum of its parts, we expanded our empirical analysis significantly.
> > >
> > > In the original draft, we initially explored metric combinations in **Appendix B.8**, analyzing a hybrid score of $\alpha \cdot \text{LENS} + (1-\alpha) \cdot \text{MDM}$. We found that shifting weight toward LENS resulted in a monotonic decrease in correlation, suggesting those two specific metrics (representation-based vs. diversity measure) do not scale synergistically.
> > >
> > > To systematically identify combinations that do yield synergistic new knowledge, we performed a comprehensive grid search combining all 5 proxies. We evaluated linear combinations of the form $S_{\text{combined}} = \sum_{i=1}^5 w_i M_i$, testing normalized weights from 0.0 to 1.0 at 0.2 intervals, resulting in an exhaustive search of $6^5$ (7,776) unique combinations. Crucially, we expanded this sweep to include both the Sentiment Analysis and Text2SQL domains to analyze how hybrid performance scales with task complexity.
> > >
> > > This expanded analysis revealed several new insights regarding how these metrics interact:
> > > - **Accurate Dataset Ranking Requires Synergy**: While individual metrics establish a baseline, combining them yields strict improvements, particularly for rank-ordering. In Sentiment Analysis, blending metrics increases the maximum single-proxy Spearman correlation ($\hat{\rho}$) from 0.68 to 0.72. In the more complex Text2SQL domain, the synergistic gains are significant: the hybrid approach boosts the maximum single-proxy Spearman correlation ($\hat{\rho}$) from 0.65 to 0.89, and the maximum Pearson correlation ($\hat{r}$) from 0.68 to 0.79.
> > > - **Task Complexity Dictates the Optimal Blend**: The optimal hybrid composition shifts significantly depending on the nature of the task. For simple stylistic tasks like Sentiment Analysis, representation-based metrics dominate the top combinations, with MDM acting as a primary anchor. However, for structurally complex reasoning tasks like Text2SQL, LENS becomes a critical component for both Pearson and Spearman correlations, consistently requiring heavy weights (between 0.6 and 1.0) to achieve optimal utility prediction.
> > >
> > > **The Core Takeaway (New Knowledge)**: The necessity of this blend confirms that *these metrics measure fundamentally distinct properties and capture orthogonal blind spots*. A synthetic dataset might exhibit excellent diversity but drift out-of-domain, or it might perfectly mimic the real distribution's style while suffering from severe mode collapse. Furthermore, standard representation metrics may miss nuanced logical errors in complex queries that LLM-based reasoning (LENS) catches. The top-performing hybrid proxies act as a counterbalance, yielding a strictly superior selection mechanism than relying on diversity alone.
> > >
> > > Finally, regarding your alternative suggestion of dissecting diversity into sub-metrics: we find this to be a highly compelling direction. While our analysis here highlights the synergistic power of combining our existing proxy suite across different task complexities, isolating different structural types of diversity (e.g., semantic vs. syntactic) is an excellent avenue for future work that we will highlight in our conclusion.
> > >
> > > We have added this comprehensive combined metric analysis to the final manuscript in **Appendix B.9** to strengthen the paper's contribution.

---

> > > > ### Comment · Reviewer_drNk · 2026-04-02
> > > > **Reviewer drNk's response**
> > > >
> > > > I once again thank the authors for their detailed response and willingness to incorporate the reviewers' feedback. I am satisfied with the contributions of this work. I wish all the luck to the authors.

---

> > > > > ### Author Response · Authors · 2026-04-02
> > > > >
> > > > > We are glad our response addressed your concerns. We sincerely thank you for your detailed feedback in strengthening the empirical depth of our work.

---

### Decision · Action_Editor_8GvK · 2026-04-27

**Recommendation:** Accept as is

**Additional Comments:**

Overall this seems like a solid paper.  The reviewers all recommended accept (with one "leaning").  They considered the synthetic dataset evaluation problem to be highly relevant and interesting, and found the methods presented by the authors as "neat".  The paper is well written, well-motivated and seems empirically thorough.  The major concern raised by the reviewers was regarding novelty compared to existing work.

**Audience:**

Yes

**Audience Explanation:**

The reviewers noted that this is a timely problem that is relevant to the community.  Synthetic datasets are of a lot of interest right now, as large models are exhausting existing real datasets and data has become a bottleneck.

**Claims And Evidence:**

Yes

**Claims Explanation:**

The consensus of the reviewers was that the claims were substantiated by the evidence provided.  In particular, the reviewers praised the "comprehensive" benchmarking across multiple domains.  One reviewer was concerned about the metrics used and another brought up a weakness of MDM in particular, but they seem to have been convinced by the author response.  Another reviewer found that the statistical strength of the experiments (pearson correlation scores) were rather weak, though the authors also seem to have addressed that concern in their revisions.